# Cognition as Morphological/Morphogenetic Embodied Computation In Vivo

**DOI:** 10.3390/e24111576

**Published:** 2022-10-31

**Authors:** Gordana Dodig-Crnkovic

**Affiliations:** 1Department of Computer Science and Engineering, Chalmers University of Technology, 41296 Gothenburg, Sweden; gordana.dodig-crnkovic@chalmers.se; 2Division of Computer Science and Software Engineering, School of Innovation, Design and Engineering, Mälardalen University, 722 20 Västerås, Sweden

**Keywords:** embodied cognition, evolution, agency, autonomy, intelligence, morphological computing, morphogenesis, natural computing, computation, information

## Abstract

Cognition, historically considered uniquely human capacity, has been recently found to be the ability of all living organisms, from single cells and up. This study approaches cognition from an info-computational stance, in which structures in nature are seen as information, and processes (information dynamics) are seen as computation, from the perspective of a cognizing agent. Cognition is understood as a network of concurrent morphological/morphogenetic computations unfolding as a result of self-assembly, self-organization, and autopoiesis of physical, chemical, and biological agents. The present-day human-centric view of cognition still prevailing in major encyclopedias has a variety of open problems. This article considers recent research about morphological computation, morphogenesis, agency, basal cognition, extended evolutionary synthesis, free energy principle, cognition as Bayesian learning, active inference, and related topics, offering new theoretical and practical perspectives on problems inherent to the old computationalist cognitive models which were based on abstract symbol processing, and unaware of actual physical constraints and affordances of the embodiment of cognizing agents. A better understanding of cognition is centrally important for future artificial intelligence, robotics, medicine, and related fields.

## 1. Introduction

Currently, cognitive science is a field in transition, as a result of accumulated knowledge from a variety of constituent and closely related research fields. From the theory of science point of view, this process of paradigm shift can be understood in terms of shifting relationships between networks of theories, rather than the battle in which one theory will win. Contemporary understanding of science involves clusters of theories used as a toolbox for approaching the given research field [1]. Thus, we do not have one single theory of cognition, as we do not have one single theory of life. Even in sciences studying far simpler objects, we have clusters (networks) of theories—such as in physics, information theory (with not only Shannon’s theory of communication but a variety of semantic information theories), or theories of computation (apart from the Turing machine model, which is applicable to symbolic, sequential information processing, there are distributed concurrent and resource-sensitive computational models suitable for modeling complex systems).

Yet, there is an implicit idea about the necessity of the existence of one theory of cognition, even though by now we know that cognition is a complex process that in living beings unfolds on several levels of an organization, from cells up. A problem in many philosophical debates, including those in the philosophy of cognitive science, the philosophy of cognition, the philosophy of mind, the philosophy of information, the philosophy and theory of computation, etc., is that different theories start from different assumptions and look at different aspects of cognition. The only way to reconcile competing approaches is by looking at controversies from a higher level of abstraction, explicating their underlying assumptions. That is how we proceed in this article and propose info-computation as a possible unifying approach, without the intention to claim that it is the only correct, right, or only appropriate perspective. We link together seemingly unrelated or even apparently contradicting research results and show how they make sense in a broader context. The current state of the art is like a kaleidoscope configuration—rotate the whole (change the perspective, framing, or focus) and you may find interesting new connections, patterns, meanings, and applications.

This article consists of the following sections:The established view of cognition and its open problems;New developments in cognitive science and contributing fields;Foundations of cognition: self-organization and autopoiesis;Cognition as a driver of evolution and evolution as a driver of cognition in living organisms and their relation to the extended evolutionary synthesis;Morphological/morphogenetic computation:
5.1.Morphogenesis as morphological computation generating an organism from active matter (embryogenesis, development, evolution);5.2.Cells as information-processing agents. Morphogenesis as Bayesian inference. Free energy principle, agency;
Conclusions.

## 2. The Established View of Cognition and its Open Problems

In The Stanford Encyclopedia of Philosophy Thagard defines cognitive science in the following way:

”In its weakest form, cognitive science is just the sum of the fields mentioned: psychology, artificial intelligence, linguistics, neuroscience, anthropology, and philosophy. Interdisciplinary work becomes much more interesting when there is theoretical and experimental convergence on conclusions about the nature of the mind. For example, psychology and artificial intelligence can be combined through computational models of how people behave in experiments. The best way to grasp the complexity of human thinking is to use multiple methods, especially psychological and neurological experiments, and computational models. Theoretically, the most fertile approach has been to understand the mind in terms of representation and computation” [2].

The same author proposes a wider definition in the Encyclopaedia Britannica, extending “thinking” to include emotional experience: “The term cognition, as used by cognitive scientists, refers to many kinds of thinking, including those involved in perception, problem solving, learning, decision making, language use, *and emotional experience*” [3]. This extension bridges some of the distance between cognition as thinking and its underlying processes. Yet, this view of cognition is still human-centric, descriptive, and lacking *generative mechanisms* that can dynamically overarch the chasm between matter and mind from abiogenesis through the development and evolution of more and more complex cognitive systems. Thagard’s definitions of cognitive science do not mention biology, chemistry, (quantum, nano, etc.) physics or chaos theory, self-assembly/self-organization, artificial life or computing/computability and information processing, extended mind, distributed cognition studied with help of network science, sociology, or ecology and most importantly, evolution and development. All those research fields actively contribute to the current understanding of cognition. Thagard’s definition of cognition is about *high-level processes* remote from the physical–chemical–biological substrate [4]. It is modeled by classical sequential computation, that is symbol manipulation (traditional computationalism). 

Historically, behaviorism offered an alternative view of cognition with a focus on the *observable behavior* of a subject. According to [5], theoretical behaviorism comes close to the view of embodied cognition, in that it stresses the importance of behavior in contrast to the emphasis on high-level cognition typical of old cognitivism. This divide is mirrored in the present-day schism between cognitivism/computationalism and EEEE (embodied, embedded, enactive, extended) cognition. 

There have been numerous attempts to bridge this gap made by Clark, Scheutz, Pfeifer, and others [6,7,8,9,10,11,12], offering connections between sub-symbolic (signal processing) basic-level and symbolic processing, higher-level notions of cognition. 

As a result of a current narrow definition of cognition there is a long list of unsolved/unsolvable problems of cognitive science, identified by [3]:*The emotion challenge*: cognitive science neglects the important role of emotions in human thinking;*The consciousness challenge*: cognitive science ignores the importance of consciousness in human thinking;*The world challenge*: cognitive science disregards the significant role of physical environments in human thinking, which is embedded in and extended into the world;*The body challenge*: cognitive science neglects the contribution of embodiment to human thought and action;*The dynamical systems challenge*: the mind is a dynamic system, not a computational system;*The social challenge*: human thought is inherently social in ways that cognitive science ignores;*The mathematics challenge*: mathematical results show that human thinking cannot be computational in the standard sense.

## 3. New Developments in Cognitive Science and its Contributing Fields

Among the new developments in cognitive science, the most prominent is the broadening of the idea of cognition to include not only human cognition but also cognition in other living organisms. 

Early research by Lyon [13], van Duijn et al. [14], Ben Jacob et al. [15] Baluška and Levin [16], and others, is moving from an anthropocentric to a biocentric view of cognition in the search for the answer to the question “what is cognition?” One noteworthy insight is that living cells possess remarkable information-processing competencies [13,14,15,17] and that the head (brain) is not necessary for cognition [16]. In the biogenic approach, cognition is the ability of an organism to register information, interact with the environment to provide what is needed for survival, eliminate what is not needed, memorize significant experiences from the past, and act purposefully so as to increase the chances of survival of the organism, all of which can be found in a single cell.

The fundamental mechanisms of biological cognition for an organism on the border between unicellular and multicellular have been studied by Vallverdú et al. [18] for the case of slime mold, in a bottom-up approach, starting from the biological and biophysical features of the slime mold and its regulatory systems. The study builds on the Lyon’s biogenic cognition, centered on two main frameworks for understanding biology: self-organizing complex systems and autopoiesis; Muller, di Primio-Lengeler’s account of contributions of minimal cognition to the flexibility of biological agents (in comparison to the rigidity of automation technology); Bateson’s “patterns that connect” systemic view, Maturana’s autopoiesis; and Morgan’s Canon (“In no case is an animal activity to be interpreted in terms of higher psychological processes if it can be fairly interpreted in terms of processes which stand lower in the scale of psychological evolution and development.”). The authors refer to Kull’s appeal for a reformulation of the concept of cognitive process and intelligence that “requires a deeper understanding of the behavior of living systems, where cognition does not depend on the existence of a nervous system that channels it, but on functional circuits which allow the minimal perception of the surrounding environment and its biosemiotics processing in vivo [19].” An analogous claim has been made by [20], from the perspective of modeling life/cognition in terms of cognitive info-computation.

In the context of unicellular organisms with basal cognition, interesting recent attempts have been made to understand their “selfhood” as the basis of their agency. Levin studied the computational boundary of a “self” and found that developmental bioelectricity is the driver of multicellularity and scale-free cognition [21].

Based on accumulated evidence about the cognitive behaviors of unicellular organisms as well as multicellular organisms such as slime molds, far simpler than humans and other organisms with nervous systems, Fields and Levin proposed scale-free biology by integrating evolutionary and developmental thinking [22]. In a similar vein, Lyon and coauthors characterized this transition from an anthropocentric to a biocentric understanding of cognition, as “reframing cognition” and “getting down to biological basics” [23]. One of the consequences of recognizing basal cognition is the increased interest in the diverse cognitive capacities of single cells and their influence on the macroscopic cognition of an organism, such as, e.g., Verny’s study of the mechanisms of cellular memory [24].

A significant contribution to the establishment of the field of basal cognition was made in a Special Issue of *Philosophical Transactions B*, titled “Basal cognition: multicellularity, neurons and the cognitive lens”, presenting this major novel direction of research on cognition [25]. An important link between traditional neural and newly recognized basal cognition is made through morphological coordination of information processing which was found to be a common inherited function connecting neural and non-neural signaling [26].

The controversy about computational (high-level, symbol-based) vs. embodied (low-level, signal-based) cognition, has already been addressed in the book *Computationalism new directions* [9] and is still highly relevant. The main message of the book was that not all kinds of computations are symbol manipulation and that new computationalism is capable of considering embodiment, signal processing, and interactions of the cognizing agent with the environment.

The recent symposium “Rethinking Computational Approaches to the Mind. Fundamental Challenges and Future Perspectives” addressed problems with old computationalism and introduced new computationalist approaches [27]. Linzier and Polany confronted the issue of *supposed contradiction between computational and dynamic models* and presented an information–theoretic toolkit for studying the dynamics of complex systems. [28]. Brette offered a critique of the approaches to *computation in the brain seen as computation over neural representations,* where neural representations are supposed to correspond to neurophysiological states, which are then assumed to be governed by computational processes, arguing that neural representations are not states of the brain [29]. Varoquaux pointed out the current difficult position of brain sciences because of *the abundance of theories, and data on the brain and behavior*, and the lack of a framework to connect and interpret them. His advice is to “focus on models that generalize across many experimental settings and criticize models more on their predictions than on their ingredients.”. Flack presented the view of *collective computation in nature with distributed concurrent information processing in biological systems*, shaping the foundations of computation and micro–macro relations in nature. Finally, Mitchell suggested that the historical succession of “AI springs” and “AI winters” is a result of our *insufficient understanding of the nature and complexity of intelligence*. In a nutshell, the symposium adds novel research insights and contributes to a more nuanced and interdisciplinary science-based understanding of cognition than what was the case in the time of symbol-based disembodied old computationalism.

## 4. Foundations of Cognition: Self-Organization and Autopoiesis

Recognizing the cognitive capacities of every living cell has already been established by Maturana and Varela [30] and Stewart [31], who argued that *cognition should be seen as a realization of life*, thus as biocentric. In Stewart’s formulation, cognition = life, so consequently, it must be seen in the light of evolution, as “nothing in biology makes sense except in the light of evolution” Dobzhansky [32], the topic is addressed by Miller, Torday, and Baluška [33].

Therefore, if we want to understand cognition, we must search for its roots and answer the question: how did life start?

Life is characterized by the process of autopoiesis, and self-(re)creation [30]. However, simpler capacities of self-assembly and self-organization exist also in inanimate matter.

Already Haken in 1985 [34] studied self-organization in physics, while Tiezzi in his book *Towards an Evolutionary Physics* described the evolution on the level of physics, explaining that the self-organization of a physical system is made possible by the presence of a boundary of a system. Boundary serves as an interface that allows the control of exchanges between the system and the environment. This process conditions its evolution, [35].

The next level of self-organization, chemistry, was described by Orlik in his “Introduction to self-organization in chemical and electrochemical systems” [36].

In a further step, chemical evolution is leading to the emergence of life. It was a topic of the Special Issue of the journal *Chemical Reviews* exploring the chemical origins of life, and thus cognition [37]. In search of a chemical basis for minimal cognition Hanczyc and Ikegami studied a simple chemical system consisting of an oil droplet in an aqueous environment and found that: “a chemical reaction within the oil droplet induces an instability, the symmetry of the oil droplet breaks, and the droplet begins to move through the aqueous phase.” Authors found indications of the presence of feedback cycles capable of forming the basis for self-regulation, homeostasis, and perhaps also autopoiesis. They discuss the result that “simple chemical systems are capable of sensory-motor coupling and possess a homeodynamic state from which cognitive processes may emerge”, [38].

Crucial steps to life, from chemical reactions to code-using agents, have been analyzed by Witzany who addressed the question: “How did prebiotic chemistry start life?”, [39].

Further on, following abiogenesis, newly formed pre-biotic agents started to evolve. In the book *Self-organization as a New Paradigm in Evolutionary Biology: From Theory to Applied Cases in the Tree of Life*” [40], the author draws together contributions of diverse strings of self-organization to evolutionary biology.

In the hierarchy of complexity, the next level above individual organisms are communicating groups, which have been examined by Heylighen, who observed the emergence of coordination, shared references, and collective intelligence in such groups [41]. On the level of distributed cognition, the work of Magnani adds a new perspective of eco-cognitive computationalism, following the development; from the mimetic minds to the morphology-based enhancement of mimetic bodies [42].

In sum, the emergence of new increasingly complex cognitive organisms from the simpler ones progresses through the processes of self-assembly and self-organization as a new paradigm in evolutionary biology [40], and autopoiesis [30,43], by way of continual interactions of constituent parts with each other and with the environment. This view has been supported by the evidence in the recent research literature. From those empirical, experimental, and theoretical results it follows that understanding cognition requires an updated generalized model of evolution from physics to chemistry, biology, and cognition.

## 5. Cognition as a Driver of Evolution and Evolution as a Driver of Cognition in Living Organisms. Relation to the Extended Evolutionary Synthesis

A new, more general view of cognition that recognizes not only humans as cognitive agents but includes all living beings, casts a completely new light on the process of evolution. Evolution has been described as a cognition-based process driven by non-random genome editing and natural cellular engineering [44]. Torday and Miller elaborated on the view of cognition-based evolution in the context of epigenetic evolutionary biology [45].

”Cognition-Based Evolution argues that all of the biological and evolutionary development represents the perpetual auto-poietic defense of self-referential basal cellular states of homeostatic preference. The means by which these states are attained and maintained is through self-referential measurement of information and its communication” [44].

Cellular and evolutionary perspectives on organismal cognition, from unicellular to multicellular have been studied by Baluška, Miller, and Reber [46] who have been following the evolutionary origins of cells as unicellular organisms and their evolution to multicellularity, based on two fundamental self-identities (also found in humans): the immunological and the neuronal.

Arguments for biological evolution as a defense of “self” are presented by [33] who found that ”As self-referential cognition is demonstrated by all living organisms, life can be equated with the sustenance of cellular homeostasis in the continuous defense of “self”.

The connectionist approach to the evolutionary transitions in individuality is the subject of study of the recent EVO-EGO project [47].

Accumulated newly acquired research results, among others about the role of self-organization, the role of interactions with the environment and cellular evolution, lead Laland, Uller, and Feldman et al. to ask the question: “Does evolutionary theory need a rethink?” [48]. The authors provide evidence for an affirmative answer. In the years to come, we can expect further developments in the new evolutionary synthesis, incorporating a growing body of research results on the evolution on the micro-level.

Apart from the proposal for an update of evolutionary theory motivated by new empirical data about the behavior of microorganisms, there was already a proposal in 2014 to view evolution in four dimensions: genetic, epigenetic, behavioral, and symbolic [49], based on models of cognitive learning. The recent related work of Ginsburg and Jablonka studies specifically the evolution of the “sensitive soul” [50], a concept borrowed from Aristotle. In Aristotle’s view, plants have a nutritive soul, animals have a sensitive soul, and human beings have a rational soul. A soul, as elaborated in Aristotle’s *De anima* is “*the actuality of a body that has life*,” where life is the capacity for self-sustenance, growth, and reproduction. Aristotle’s formulation is close to Maturana and Varela’s notion of cognition as *autopoiesis*, i.e., ”realization of the living” [30]. Of central importance for Ginsburg and Jablonka is the interaction and the communication between an agent and its environment, which changes both. They point out the central role of the interplay between the genetic code and the environment, through material embodiment, as fundamental for evolution.

Evolution can be understood as a learning process, in which living agents learn through interactions with their environment. Watson, Richard, and Szathmary [51] pose the question: how can evolution learn? and propose the answer: “Specifically, connectionist models of memory and learning demonstrate how simple incremental mechanisms, adjusting the relationships between individually simple components, can produce organizations that exhibit complex system-level behaviors and improve the adaptive capabilities of the system. We use the term “*evolutionary connectionism*” to recognize that, by functionally equivalent processes, natural selection acting on the relationships within and between evolutionary entities can result in organizations that produce complex system-level behaviors in evolutionary systems and modify the adaptive capabilities of natural selection over time” [52].

This process of learning can be expressed in computational form. In terms of relationships between living agents and their environment, morphological/natural/physical computation presents a basic mechanism for the process of learning as argued in [53].

## 6. Morphological/Morphogenetic Computation

Novel developments in computational approaches to cognition and intelligence [54], as well as robotic implementations [11,55], show that the body is an integral part of cognitive processes, connecting data to the agency. Computation is not only symbol manipulation but also physical processes in the body of the cognizing agent known as morphological computation/natural computation/unconventional computation/physical computation, and similar, [56].

In the framework of computing nature/info-computationalism, information is always embodied in the physical substrate, and represents the world for an agent, while physical/embodied/morphological computation stands for the dynamics processes unfolding in informational structures. Life and thus cognition is a necessary consequence of the properties of physical matter that has self-organized, starting with physical and chemical abiogenesis that has led to the emergence of living creatures. They continued evolving driven by their own internal agency.

Morphology is defined by the shape/structure and material properties of an agent (a living organism or a machine in the case of robotics) that enables and constrains its possible interactions with the environment as well as its development, including its growth and reconfiguration. The evolution of living organisms explained info-computationally is different from the classical view of evolution based on random mutations + selection. An info-computational approach based on the information and (morphological) computation models goal-directed sub-organismic (physical and chemical) and organismic (biological) agency at a variety of levels of an organization. Mutations are governed not only by chance but also by self-organizing properties of active matter, the morphology, and the intrinsic agency of a living organism as a thermodynamical system far from equilibrium. This view of evolution is compatible with the modern extended evolutionary synthesis [48], with its constructive development and reciprocal causation which is a consequence of the interaction of a biological agent with the environment.

In the article ”Information, Computation, Cognition. Agency-based Hierarchies of Levels” [57] the author provides arguments for the new kind of understanding, in the sense of Wolfram [58], of lawfulness in the organization of nature and especially living systems, emergent from generative computational laws of self-organization based on the concept of agency. In order to understand the world, the organization of the parts in the whole and interactions between them are central, and generative processes such as the self-organization [59] (which acts in all physical systems), along with autopoiesis [30] (that acts in the living cell as a whole). Morphological computation is presented as the self-organization mechanism of a cognizing agent [60].

From the computational-theoretic point of view, self-organization and autopoiesis can be described by agent-based models, such as the actor model of computation [61] which is suitable for modeling concurrent physical processes. The actor model of computation builds on the exchange of information (messages) between agents (actors). It is directly related and can be expressed in terms of signal communication between agents, which is elaborated by Skyrms in his book *Signals: Evolution, Learning, and Information*, [62].

### 6.1. Morphogenesis as Morphological Computation Generating an Organism from Active Matter (Embryogenesis, Development, Evolution)

The recent trend of “*a reattachment of form to matter*” emphasizes an understanding of “information” as inherently embodied. It was analyzed by Keller, who noticed that in chemistry, cognitive science, molecular computation, and robotics one is turning to biological processes for an establishment of a new epistemology, [63]. Lehn observed the same trend in chemistry, claiming that the vision of supramolecular chemistry is a “general science of *informed matter”*, promoting in chemistry the third component (information) in the basic triplet of matter-energy-information, [64].

Roots and promises of chemical-based computing (i.e., pseudo biological information processing) based on general principles of information processing by chemical-based biomolecular systems able to solve effectively problems of high computational complexity, have been characterized by Rambidi as including: “very large-scale parallelism of information processing, high behavioral complexity, the complementarity of information features, self-organization, and multilevel architecture” [65].

Taking into account that cognitive bio-centric systems can be viewed at a succession of levels of abstraction, from molecular to organismic and social/collective, Hogeweg observed “the importance of interactions between processes at diverse space and time scales in biological information processing”, also presents as a recurrent theme in the work of Michael Conrad [66].

More details on the current state of research on mechanisms of processes of morphogenesis, the genesis of forms, may be found in Pismen’s book *Morphogenesis Deconstructed. An Integrated View of the Generation of Forms*, where the study of connections between morphogenetic processes in inorganic, biological, and social systems has been presented [67].

### 6.2. Cell Seen as an Information Processing Agent—Morphogenesis as Bayesian Inference: Free Energy Principle, Agency

Starting from the insight that ”morphogenesis could be understood in terms of cellular information processing and the ability of cell groups to model shape” [68] offers a proof of principle that self-assembly is an emergent property of cells that share a common (genetic and epigenetic) model of organismal form. This allows the authors to interpret a system as “inferring the causes of its inputs—and acting to resolve uncertainty about those causes.” This novel perspective balances strategies with a focus on molecular pathways and bottom-up causation, with top–down constraints found in cybernetics and neuroscience. The driving force in this process is the minimization of surprise/prediction error. Friston et al. [68] find that by using a variational free energy approach to pattern formation and control in complex biological systems, *the process of morphogenesis can be interpreted as Bayesian inference.*

Kuchling et al. specify “in neuroscience, a variational free energy principle has been used, with a formulation of self-organization in terms of active Bayesian inference. The free energy principle has also been applied to biological self-organization in somatic cells, for processes of development or regeneration. The Bayesian inference framework treats cells as information processing agents, where the driving force behind morphogenesis is the maximization of a cell’s model evidence” [69].

Fields, Friston, Glazebrook, Levin, and Marciano [70] focused on the neuromorphic development understood as a result of the free energy principle. They argue that “any system with morphological degrees of freedom and locally limited free energy will, under the constraints of the free energy principle, evolve toward a neuromorphic morphology that supports hierarchical computations in which each “level” of the hierarchy enacts a coarse-graining of its inputs, and dually a fine-graining of its outputs. Such hierarchies occur throughout biology, from the architectures of intracellular signal transduction pathways to the large-scale organization of perception and action cycles in the mammalian brain” [70].

Consequently, the free energy principle in pattern formation and morphogenesis provides a quantitative formalism for understanding cellular decision-making in the context of embryogenesis, regeneration, and cancer suppression. Fields et al. [70] illustrate this ‘first principle’ approach to morphogenesis in the case of planarian regeneration showing that simple modifications of the inference process can affect patterning in developmental and regenerative processes without changing the DNA. This approach to understanding the self-organized behaviors of embodied agents as satisfying basic constraints of sustained interactions with the environment enables new models of developmental change in evolution.

As life-like behaviors emerge in coupled dynamical systems, Friston used simple simulations to study this process unfolding through perception and action. The conclusion is that for an agent to exist in a changing world, it has to model that world and infer the causes of its sensations, which is achieved by active inference [71]. Isomura explains how Friston’s free-energy principle accounts for biological organisms’ perception, learning, and action. The dynamics of neural networks minimize cost functions and can be modeled in terms of the free-energy principle. This equivalence allows us to explain network dynamics and predict subsequent learning [72].

Badcock, Friston, and Ramstead proposed a unifying theory of the embodied, situated human brain across different spatiotemporal scales, called the Hierarchically Mechanistic Mind (HMM) as the hierarchically mechanistic mind with a free-energy formulation of human cognition. HMM combines the free-energy principle in neuroscience with an evolutionary systems theory of the brain, cognition, and behavior relating to Tinbergen’s four questions: function/adaptation, evolution/phylogeny, development/ontogeny, and causation/mechanism [73].

Pio-Lopez, Kuchling, Tung, Pezzulo, and Levin applied active inference and morphogenesis, to computational psychiatry. Active inference as a leading theory in neuroscience provides a simple and biologically plausible account of how action and perception are coupled in leading to Bayesian optimal behavior. Morphogenesis has been described as the behavior of cellular collective cognition/intelligence. The authors found a link between cell biology and neuroscience, identifying developmental defects in morphogenesis as disorders of active inference, i.e., disorders of information processing. The authors concluded that: “*The use of conceptual and empirical tools from neuroscience to understand the morphogenetic behavior of pre-neural agents offers the possibility of new approaches in regenerative medicine and evolutionary developmental biology*” [74].

Moving in the same direction “a primer on conceptual tools for analyzing information processing in developmental and regenerative morphogenesis” gives a perspective seen through “the cognitive lens” [75].

In the context of human cognition, closely related to the Bayesian brain framework is the predictive coding theory which offers a potentially unifying account of cortical function—“*postulating that the core function of the brain is to minimize prediction errors with respect to a generative model of the world*”. In his take on the issue, Clark connects predictive brains, situated agents, and the future of cognitive science [76]. For a theoretical and experimental review of predictive coding of the brain, see [77].

## 7. Conclusions

In the Introduction section of this article, we pointed out that there is a long list of unsolved/unsolvable open problems in cognitive science, which are the result of a currently prevailing narrow definition of cognition, identified by Thagard [3]. New research results support a broader understanding of cognition as the capacity of all living organisms, which can be modeled as information processing (morphological/morphogenetic computation) enabling new solutions to the existing challenges. In what follows, we revisit the key challenges and some additional challenges that result from the current synthesis:
*The emotion challenge*: Currently, cognitive science neglects the important role of emotions in human thinking. This challenge is being met on various fronts. Recent research shows how emotions, feelings, and sensations are a result of embodied information processing and are inseparably related to other computational cognitive processes, see [78,79,80,81]. In the Bayesian perspective, with a focus on human cognition, and based on the free energy principle, the emotional inference is an active area of research from two key directions. First, as interoceptive inference [82,83,84,85]. Second, the emotional valences of various belief states are viewed through the lens of resolving uncertainty as the mathematical image of angst and anxiety, [86,87].*The consciousness challenge*: Currently, cognitive science ignores the importance of consciousness in human thinking. Several information–theoretic approaches are being developed to address consciousness. For example, the information integration solution, see [88,89,90,91]. The information geometry approach that builds on a formal specification of the boundary between a living system and its external environment (a Markov blanket) [92] has been introduced. Within a framework of dual-aspect monism, intrinsic and extrinsic information geometry are providing the link between the brain and mind [93,94].*The world challenge*: Currently, cognitive science disregards the significant role of physical environments in human thinking, which is embedded in and extended into the world. This represents the main motivation for the pragmatic turn and an activist or situated approach to cognition. The circular causality between a bounded cognition itself and the world is central for active inference and learning; in which the agent is observing sensations from her world. In machine learning, the role of a generative model has been introduced through the notion of a world model [95]. See also [96] for an elaboration of the path to physics from computing [97] on the process from information to behavior [98] on computation in neuroscience, and [54,60] with arguments for mechanisms of morphological computing as reality construction for a cognizing agent. As the development of artificial intelligent cognitive computational systems progresses, a framework that can connect the natural with the artificial is used for learning in both directions—from the natural system to the artificial and back, [99]. For example, we can mention the impact of deep learning research on understanding cognition. The recent work [100] discusses the necessity of embodiment in the case of LLM (Large Language Models).*The body challenge*: Currently, cognitive science neglects the contribution of embodiment to human thought and action. See previous paragraphs on the fundamental importance of embodiment to the processes of cognition, as argued by [4,6,101,102,103].This challenge has also inspired much of the work on interoception (the perception of sensations from inside the body) noted in [104], and the close relationship between action and perception resulting from the embodied brain with active vision and sensing, which is also a current focus in artificial intelligence research.*The dynamical systems challenge*: Currently, there is a supposed contradiction between dynamics systems and (old) computationalism, between the mind conceived as a dynamical system, and a computational system. A broader understanding of computation that includes dynamical systems solves this apparent contradiction. For the arguments from the theory of computation, see [105,106].The free energy principle addresses this challenge by developing a physics of sentience combining dynamical systems theory with the boundary separating self from nonself. Coupling of the dynamics of the particular partition of states external and internal to a system to the corresponding information geometry of belief updating and inference is carried out by Bruineberg et al. [107]*The social challenge*: Human thought is inherently social in ways that cognitive science ignores. Info-computational studies of social cognitive systems already exist that extend cognition to groups of cognizing agents, [22,41,46,108,109]. In a Bayesian setting, social cognition is addressed through interpersonal inference and niche construction as enactive and distributed inference. These lines of inquiry range from the nature of dyadic interactions to the spread of ideas over communities, [110,111].*The mathematics challenge*: Mathematical results show that human thinking cannot be computational in the standard sense, as shown by Cooper [6,7,8,9,54,76,112,113]. In addition, a move from inductive and deductive logic to the perspective of active inference brings forward abductive reasoning [114], which is making the best guess about the external states of affairs [93,115]. The key question here is the distinction between dynamics and belief updating on continuous states as opposed to discrete state space models used for a symbolic representation. This has emerged in the distinction between predictive processing (under continuous state space models) such as predictive coding [116,117], relative to the use of belief propagation and variational message passing (under discrete state space models).


To this list of open problems with the current view of cognition made by references [3,4], we can add the following based on the recent research results on embodied cognition and new computationalism:
8.*The computational architecture challenge*: Cognition is not only the result of the activity of the brain, not even the activity of neurons alone. It is the capacity of all somatic cells, that are interacting with each other, and with the environment, [23,25]. This challenge arises in many guises in different fields. For example, in radical constructivism, it is known as structure learning [118,119]. In Bayesian formulations of active inference, it is reduced to Bayesian model selection, which may be the mathematical image of natural selection [120]. In other words, evolution itself may be a belief-updating process in which the likelihood of various phenotypes reflects their fit to the environment as scored by things such as variational free energy [121].9.*The generative mechanisms challenge*: Mechanistic models of cognition provide generation of cognition through the processes of (morphological/morphogenetic) computation, unfolding in networks of agents from molecules to biological organisms. Cognition is first understood when we know its generative mechanisms (constructive approach), [24,31,71]. This is an active field of research that in Bayesian learning, focuses on the structure and form of generative models that underwrite active inference and learning—and the nature of message passing that is realized in terms of biophysics.10.*The information processing (Bayesian learning) challenge*: How is evolution learning? A variational free energy principle can be used, to formulate self-organization (morphogenesis) in terms of active Bayesian inference. In the Bayesian inference framework, cells are information processing agents, and the driving force behind morphogenesis is the maximization of a cell’s model evidence, [69,72,122,123,124].


In brief, in this paper, we compared the current view of cognition as presented in the leading encyclopedia with the perspective of the newest results of research on cognition in nature. Special attention was paid to basal cellular cognition and its connection to morphogenesis and evolution. This new approach to cognition has the potential to impact, among others, the following features:
Our fundamental understanding of cognition in nature;An understanding of mechanisms of evolution and development;The design and engineering of cognitive computational artifacts;Medical applications.


In perspective, an emerging, broader cognitive science with new computationalism as a basis, promises to contribute to important future advances in both fundamental sciences and applications.

## Data Availability

Not applicable.

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
