# Peer review of "Cognition as Morphological/Morphogenetic Embodied Computation In Vivo"

_entropy, 2022, doi:10.3390/e24111576_

Round 1

Reviewer 1 Report

I enjoyed reading this scholarly and informed overview of different approaches to sentient behaviour, cognition, computing and self-organisation in a situated context. My only suggestion is to unpack the conclusion, so that people can see how some of the key challenges are currently being addressed. In other words, you could unpack the material in square brackets under each heading and preview your final list with the following:

“In what follows, we will revisit the key challenges (and some additional challenges that emerge from the current synthesis). I will highlight recent approaches to the challenges based upon the perspectives considered above.

“The emotion challenge: current cognitive science neglects the important role of emotions in human thinking.

This challenge is being met on various fronts. For example, recent research shows how… In terms of the Bayesian perspective afforded by the free energy principle, emotional inference is now an active area of research from two key perspectives. First, the importance of gut feelings and interoception has been addressed under the rubric of interoceptive inference (Seth 2013, Fotopoulou and Tsakiris 2017, Smith, Kuplicki et al. 2020, Tschantz, Barca et al. 2022). Second, the (emotional) valences of various belief states are now viewed through the lens of resolving uncertainty as the mathematical image of angst and anxiety (Joffily and Coricelli 2013, Hesp, Smith et al. 2019).” The consciousness challenge: currently…

Several information theoretic approaches are now being brought to the table to address consciousness. For example the information processing solution…. Similarly, the information geometry that inherits from a formal specification of the boundary between a living system and its external niche (a.k.a. a Markov blanket) (Sakthivadivel 2022) has been treated under the auspices of a dual aspect monism with an intrinsic and extrinsic information geometry that may link the mind and brain (Seth 2015, Friston, Wiese et al. 2020).

The world challenge…

This probably represents the main motivation for the pragmatic turn and an activist or situated approach to cognition in the life sciences. The circular causality between a bounded itself and the world lies at the heart of active inference and learning formulations; in which the agent is responsible for soliciting sensations from her world. Interestingly, in machine learning, the role of a generative model has been foregrounded with the notion of a world model (Ha and Schmidhuber 2018).

The body challenge:…

This challenge has also inspired much of the work on interoception noted above (Barrett and Simmons 2015), and the close relationship between action and perception entailed by the embodied brain — leading to notions of active vision and sensing, which is also a current focus in artificial intelligence research.

The dynamical systems challenge….

Interestingly, the free energy principle addresses this challenge head on by developing a physics of sentience by combining dynamical systems theory with the boundary separating itself from nonself. Much of the heavy lifting then rests upon coupling the dynamics of the ensuing (particular) partition of states that are external and internal to a system to the accompanying information geometry of belief updating and inference (Bruineberg, Dolega et al. 2021).

The social challenge…

This is currently being met in a Bayesian setting through interpersonal inference and niche construction as an enactive and distributed kind of inference. These lines of enquiry range from the nature of dyadic interactions through to the spread of ideas over communities (Veissiere, Constant et al. 2019, Albarracin, Demekas et al. 2022).

The mathematics challenge.…

In addition, a move from inductive and deductive logic to the perspective afforded by active inference brings abduction to the table. In other words, making the best guess about the external states of affairs (Seth 2014, Seth 2015). Key questions that arise in this setting speak to the distinction between dynamics and belief updating on continuous states (some time), as opposed to discrete state space models that are more apt for a symbolic kind of representation. Technically, this has emerged in the distinction between predictive processing (under continuous state space models) such as predictive coding (Mumford 1992, Lee and Mumford 2003), relative to the use of belief propagation and variational message passing (under discrete state space models).

The computational architecture challenge…

This challenge arises in many guises in different fields. For example, in radical Constructivism, it is known as structure learning (Gershman and Niv 2010, Tervo, Tenenbaum et al. 2016). In Bayesian formulations of active inference, it reduces to Bayesian model selection, that may be the mathematical image of natural selection (Vanchurin, Wolf et al. 2022). In other words, evolution itself may be a belief updating process in which the likelihood of various phenotypes reflect their fit to the environment as scored by things like variational free energy (Campbell 2016).

The generative mechanisms challenge: …

In short, this is an active field of research that focuses on the structure and form of generative models that underwrite active inference and learning – and the nature of message passing that is realised in terms of biophysics.

The information processing (Bayesian learning) challenge:…"

I think you have the last challenge already covered nicely.

I hope that these suggestions help should any revision be required.

Albarracin, M., D. Demekas, M. J. D. Ramstead and C. Heins (2022). "Epistemic Communities under Active Inference." Entropy (Basel) 24(4). Barrett, L. F. and W. K. Simmons (2015). "Interoceptive predictions in the brain." Nat Rev Neurosci 16(7): 419-429. Bruineberg, J., K. Dolega, J. Dewhurst and M. Baltieri (2021). "The Emperor's New Markov Blankets." Behav Brain Sci: 1-63. Campbell, J. O. (2016). "Universal Darwinism As a Process of Bayesian Inference." Front Syst Neurosci 10(49): 49. Fotopoulou, A. and M. Tsakiris (2017). "Mentalizing homeostasis: The social origins of interoceptive inference." Neuropsychoanalysis 19(1): 3-28. Friston, K. J., W. Wiese and J. A. Hobson (2020). "Sentience and the Origins of Consciousness: From Cartesian Duality to Markovian Monism." Entropy (Basel) 22(5): 516. Gershman, S. J. and Y. Niv (2010). "Learning latent structure: carving nature at its joints." Curr Opin Neurobiol 20(2): 251-256. Ha, D. and J. Schmidhuber (2018) "World Models." arXiv:1803.10122. Hesp, C., R. Smith, T. Parr, M. Allen, K. Friston and M. Ramstead (2019). "Deeply felt affect: the emergence of valence in deep active inference." PsyArXiv. Joffily, M. and G. Coricelli (2013). "Emotional valence and the free-energy principle." PLoS Comput Biol 9(6): e1003094. Lee, T. S. and D. Mumford (2003). "Hierarchical Bayesian inference in the visual cortex." J Opt Soc Am A Opt Image Sci Vis 20(7): 1434-1448. Mumford, D. (1992). "On the computational architecture of the neocortex. II. The role of cortico-cortical loops." Biol Cybern 66(3): 241-251. Sakthivadivel, D. A. R. (2022) "Weak Markov Blankets in High-Dimensional, Sparsely-Coupled Random Dynamical Systems." arXiv:2207.07620. Seth, A. (2014). The cybernetic brain: from interoceptive inference to sensorimotor contingencies. MINDS project. Metzinger, T; Windt, JM, MINDS. Seth, A. K. (2013). "Interoceptive inference, emotion, and the embodied self." Trends Cogn Sci 17(11): 565-573. Seth, A. K. (2015). Inference to the Best Prediction. Open MIND. T. K. Metzinger and J. M. Windt. Frankfurt am Main, MIND Group. Smith, R., R. Kuplicki, J. Feinstein, K. L. Forthman, J. L. Stewart, M. P. Paulus, i. Tulsa and S. S. Khalsa (2020). "A Bayesian computational model reveals a failure to adapt interoceptive precision estimates across depression, anxiety, eating, and substance use disorders." PLoS Comput Biol 16(12): e1008484. Tervo, D. G. R., J. B. Tenenbaum and S. J. Gershman (2016). "Toward the neural implementation of structure learning." Curr Opin Neurobiol 37: 99-105. Tschantz, A., L. Barca, D. Maisto, C. L. Buckley, A. K. Seth and G. Pezzulo (2022). "Simulating homeostatic, allostatic and goal-directed forms of interoceptive control using active inference." Biol Psychol 169: 108266. Vanchurin, V., Y. I. Wolf, M. I. Katsnelson and E. V. Koonin (2022). "Toward a theory of evolution as multilevel learning." Proc Natl Acad Sci U S A 119(6): e2120037119. Veissiere, S. P. L., A. Constant, M. J. D. Ramstead, K. J. Friston and L. J. Kirmayer (2019). "Thinking through other minds: A variational approach to cognition and culture." Behav Brain Sci 43: e90.

Author Response

I would like to thank the  reviewer for insightful, detailed and constructive review  of my article. It  improved  considerably the conclusions, especially from the perspective of Bayesian approaches to EEEE cognition.

Reviewer 2 Report

The manuscript "Cognition as morphological/morphogenetic embodied computation in vivo" by Gordana Dodig-Crnkovic represents a review on a variety of fields related to biological cognition and its confluence with fundamental disciplines of natural science and computer science. Obviosusly, at the time being, there is no possiblity of articulating a plausible synthesis around biological cognition, so the author's strategy of bringing into focus a list of unsolved problems identified by Thagard (2013), first in the Intro and later in the Conclusions, contributes to give coherence to the multiple contents of the manuscript.

Perhaps the large bibliography could have been shorter, and more space devoted to discussion, although this is the author's prerrogative choice. As an aside, The reference of Kull et al (2009) does not appear in the final list (line 174), neither Van Dijk et al (2008), line 547. Curiously I checked for very vew references, and these two were missing. So, a careful checking looks necessary.

A very interesting work The biocognitive spectrum: biological cognition as variations on sensorimotor coordination,  by Marc Van Duijn (2011) was not cited. This work, a PhD Thesis, proposes a few fundamental principles of biocognition. These interesting proposals would deserve some comment in this review.

Additionally, the lack of any reference to cellular signaling systems (the term 'signal' or 'signaling' appear only twice in the text) is surprising. Aren't signaling systems of eukaryotes (and prokaryotes) fundamental tools for differentiation and morphology in multicellulars and in the multiple physiological & communication functions of organisms? So, a fundamental staple of biocognition--at least they would deserve a few comments in this tight review.

Reviewer 3 Report

This is a very informative review covering a wide range of cognitive topics. The bibliography is extensive and up-to-date. One complaint I have is that there is no comment on the impact of deep learning research, especially GANs (Generative Adversarial Networks) and LLMs (Large Language Models), on the study of cognition. LLMs are giving us new perspectives on, for example, whether language needs embodiment or not. However, it may be a bit of a challenge to get it there. If you have time, I would be happy if you can  mention it.

Reviewer 4 Report

Most of cognitive science has focused on human cognition. The authors argues persuasively that this focus is too narrow and that cognition should be approached from a biological perspective. In particular, many biological processes can be seen as performing computations resulting in some form of cognition.

I enjoyed reading this paper and don't see anything substantial to criticize. The literature discussed in this paper is very comprehensive, but perhaps the author could also have a look at alternatives to Bayesian reasoning, some in the context of agent-based models, as are for instance to be found in Douven, The Art of Abduction, MIT Press, 2022.

Author Response

I would like to thank the  reviewer for helpful literature suggestion.